Role of initial facial attractiveness in the perceived aesthetic outcome of convex profile treatment

Psomiadis Simos 1 simospsomiadis@gmail.com
Iatrou Ioannis 1
http://orcid.org/0000-0001-9686-7423 Sifakakis Iosif 2
http://orcid.org/0000-0002-3180-6272 Gkantidis Nikolaos 3 nikolaos.gkantidis@unibe.ch
1 Department of Oral and Maxillofacial Surgery, School of Dentistry, University of Athens , Athens , Greece
2 Department of Orthodontics, School of Dentistry, University of Athens , Athens , Greece
3 Department of Orthodontics and Dentofacial Orthopedics, School of Dental Medicine, University of Bern , Bern , Switzerland
Kuijpers-Jagtman Anne Marie
Electronic publication date: 2025 Oct 17
Publication date: 2025
Volume: 13
Electronic Location ID: e19997
Received 2025 Feb 26; Accepted 2025 Aug 6
Copyright: © 2025 Psomiadis et al.
Copyright year: 2025
Copyright holder: Psomiadis et al.
License: This is an open access article distributed under the terms of the Creative Commons Attribution License, which permits unrestricted use, distribution, reproduction and adaptation in any medium and for any purpose provided that it is properly attributed. For attribution, the original author(s), title, publication source (PeerJ) and either DOI or URL of the article must be cited.
License URL: https://creativecommons.org/licenses/by/4.0/

Keywords: Facial appearance, Facial attractiveness, Patient outcome assessment, Convex profile, Dental overjet, Orthodontics, Orthognathic surgery, Perception

Funding: The authors received no funding for this work.

==============================
Background

Facial attractiveness significantly influences various life outcomes, motivating individuals to seek interventions for improvement. However, the actual benefits of such interventions and patient satisfaction largely depend on the perceived changes in facial appearance. This perception may be influenced by certain mediating factors. This study aimed to investigate how pre-treatment facial attractiveness affects the perceived facial appearance changes in convex profile patients undergoing orthodontic-surgical vs. orthodontic-only treatments.

Methods

The sample comprised 36 non-growing Class II Division 1 patients, divided evenly between treatments, with similar demographics, overjet, and treatment durations. The treatments had distinct effects on the facial contour angle (orthodontics-only: −1.2 ± 2.1°, combined orthodontics/surgery: −6.2 ± 3.9°; p < 0.001). Pre-treatment attractiveness and perceived facial changes were evaluated using frontal and profile photographs, assessed on a 100 mm visual analog scale (VAS). Each photographic set was rated by 10 surgeons, 10 orthodontists, 10 patients, and 20 laypeople, resulting in 3,600 completed questionnaires for perceived treatment effects and an additional 3,600 for attractiveness ratings.

Results

The overall facial attractiveness was lower than average (mean: 36.1), with orthodontics-only patients being significantly more attractive than patients that received surgery (mean difference: 8.2%, p < 0.001) and no significant influence of patient sex or rater type. For every 1% increase in overall facial attractiveness, there was a 0.3% decrease in the perceived facial change (p = 0.002). After accounting for pre-treatment facial attractiveness, the combined orthodontic/surgical treatment led to a substantial improvement (16%) of facial appearance, as compared to the orthodontics-only group.

Conclusions

The study highlights the critical role of facial attractiveness in shaping perceived treatment outcomes. Pre-treatment facial and profile attractiveness were inversely associated with perceived changes, suggesting that patients with initially lower attractiveness experienced more noticeable improvements. After accounting for facial attractiveness, combined orthodontic and surgical treatment was perceived significantly more favorably than orthodontic treatment alone in patients with convex facial profiles. These findings underscore the importance of incorporating facial attractiveness considerations into patient counseling and treatment planning.

Introduction

Skeletal and dental imbalances are frequently observed both in the general population and among individuals seeking orthodontic or orthognathic treatment (Josefsson, Bjerklin & Lindsten, 2007). Early diagnosis of facial skeletal problems, before the pubertal growth spurt allows for growth modification strategies (Giuntini, McNamara & Franchi, 2023). Alternative treatment plans may aim at dentoalveolar or skeletal compensation with orthodontic treatment or minor surgical interventions, respectively, or at correction of the skeletal problem with major orthognathic surgery. The latter is nowadays considered a routine treatment, although it is occasionally related to complications and patients’ refrain from treatment (Hågensli, Stenvik & Espeland, 2014; Agbaje, Luyten & Politis, 2018). Growth modification is a minimally invasive, standard procedure; however, its effectiveness in significantly enhancing facial aesthetics at rest remains controversial (Koretsi et al., 2015; Zymperdikas et al., 2016; Tsiouli et al., 2017; Zouloumi et al., 2019). Unfortunately, there is no clear boundary between these treatment indications, and the choice is largely based on the perceived patient’s needs, in addition to the clinician’s experience and convictions, as well as the importance attributed to the dental or skeletal elements of the anomaly. Nowadays, machine learning models are rapidly evolving to aid decision-making (Du et al., 2024), but the baseline information and data required to treat these models are still deficient.

Orthognathic surgery improves several aspects of the patients’ quality of life. While several parameters may determine the treatment efficacy, i.e., improvements in dental function and esthetics, temporomandibular joint disorders/muscle pain or periodontal condition, social and aesthetic aspects of treatment have a profound impact (Schaefer et al., 2024). Moreover, the improvement in facial esthetics is the main reason for which patients with dentofacial deformities seek treatment and also affects largely patient satisfaction (Pachêco-Pereira et al., 2015, 2016). This seem perfectly reasonable when considering the high impact of facial attractiveness in several fundamental life outcomes (Little, Jones & DeBruine, 2011).

Substantive research has been undertaken on the perceived treatment outcomes of convex profile individuals, primarily involving the evaluation of pre- and post-treatment photographs by assessors of diverse scientific background, age, and gender (Zouloumi et al., 2019; Psomiadis et al., 2023; Conville, Benson & Flett, 2024). However, few studies comparatively evaluated orthodontics-only vs. surgical patients using actual patient photos (Proffit, Phillips & Douvartzidis, 1992; Shell & Woods, 2003; Psomiadis et al., 2023, 2025). In an earlier attempt, orthodontists and maxillofacial surgeons rated the esthetic changes from frontal and profile photographs. Orthodontics-only patients received higher pre-treatment mean rating, which remained unchanged after treatment. Surgical patients had lower pre-treatment ratings, which improved post-treatment but remained below those of orthodontics-only patients (Proffit, Phillips & Douvartzidis, 1992). However, specialists’ groups may be more generous with their improvement ratings compared to non-specialists (Ng et al., 2013; Gkantidis et al., 2013; Zouloumi et al., 2019; Kalin, Iskender & Kuitert, 2021). A study comparing aesthetic outcomes in adult Class II patients treated with orthognathic surgery and adolescents treated with growth modification found no significant differences in pre- and post-treatment scores or overall improvement. In this study, a mixed panel of raters assessed simultaneously presented frontal and lateral facial photographs, with the treatment status arranged randomly (Shell & Woods, 2003). It can be argued here that the compared groups were not similar in terms of age, and thus of dentofacial characteristics, since the adolescent patients would express their inherent growth potential towards altering facial appearance, even without treatment (Koretsi et al., 2015; Zymperdikas et al., 2016). A recent study on orthognathic surgery patients showed notable improvements in facial appearance, with an approximate 20% enhancement in actual profile photos, in contrast to the lack of effects seen with camouflage orthodontic treatment. These improvements were consistently perceived by various groups of assessors (Psomiadis et al., 2023) and were retained when the overall facial appearance was assessed in another study (Psomiadis et al., 2025).

The aforementioned studies provide conflicting evidence attributed to different methodologies and several confounding factors. Post-treatment facial attractiveness is the primary outcome, but pre-treatment attractiveness might also influence decision-making and perceived treatment benefits (Proffit, Phillips & Douvartzidis, 1992; Burk et al., 2022a; Farshidnia et al., 2023). While prior studies linked initial dentofacial severity (Shelly et al., 2000; Tsang et al., 2009) and distinct facial features, such as the nose (Cankaya, Celebi & Bicakci, 2019), to treatment perception, direct evidence on facial attractiveness influencing perceived changes remains lacking. We hypothesize that patients with higher initial facial attractiveness may experience fewer perceived changes post-treatment compared to those with lower attractiveness. To test this, we analyzed existing data on perceived facial changes in convex profile patients who underwent orthognathic or orthodontic treatment (Psomiadis et al., 2023, 2025), in combination with newly collected pre-treatment attractiveness ratings for the same individuals.

Materials and Methods

The study belongs to a series of a large project, parts of which have been published in previous reports (Psomiadis et al., 2023, 2025). Therefore, certain methodological details are repeated here to allow full comprehension of the present study by the readers. The study protocol received approval from the Research Ethics Committee of the Dental School at the National and Kapodistrian University of Athens, Greece, prior to its initiation (Approval Date: 22.06.2018, Protocol Number: 361). Written informed consent was obtained from all participants, permitting the use of their data for research purposes.

Sample

As published previously (Psomiadis et al., 2023, 2025), the sample consisted of consecutively treated patients selected from the records of the Department of Orthodontics, Dental School, National and Kapodistrian University of Athens, Greece. A retrospective sampling approach was employed, starting from June 2018 and moving backward, to identify the most recent Class II Division 1 patients with a convex facial profile. The goal was to establish two groups of 18 patients (Group A and Group B) with comparable sex distributions. Group A included non-growing patients with a convex profile treated with fixed orthodontic appliances in both jaws combined with orthognathic surgery in one or both jaws. Group B comprised non-growing patients with a convex profile treated solely with fixed orthodontic appliances. The sample size was deemed sufficient based on empirical evidence and considerations of resource availability and study feasibility (Gkantidis et al., 2013; Tsiouli et al., 2017; Zouloumi et al., 2019) and was confirmed by post hoc power analysis (Psomiadis et al., 2023, 2025). Both groups met the following eligibility criteria: Class II Division 1 pre-treatment occlusion with a molar relationship greater than half cusp Class II on both sides and an overjet between 6 and 12 mm, a convex skeletal profile (ANB angle between 5° and 9°), and a convex facial profile based on photographs (facial contour angle: 15°–25° for males, 17°–27° for females). They also had a Frankfort-mandibular plane angle (FMA) between 17.5° and 32.5°, no history of esthetic surgical soft-tissue interventions, no congenital craniofacial anomalies, syndromes, or significant facial asymmetries, and completed skeletal growth at baseline (age >15 years and CVM stage ≥5). Additionally, participants had no missing teeth other than third molars, did not discontinue treatment, and had not used mandibular advancement devices during orthodontic treatment.

Beyond the inclusion criteria, patients were selected irrespective of the specific details of their surgical or orthodontic treatment plans. Treatment procedures may vary, with plans tailored to individual clinical needs. Variations are common even among clinicians managing similar clinical conditions, and patient responses can also differ significantly. The underlying assumption of our study was that each treatment plan aimed to achieve a satisfactory improvement in function and aesthetics, as defined collaboratively by the clinician and the patient. The initial diagnostic data were utilized during sample selection, while the final diagnostic data were verified solely for availability.

Changes in facial appearance were assessed by examining facial photographs taken with the Frankfurt-Horizontal plane parallel to the ground, teeth in maximum intercuspation (light contact), and lips at rest.

The similarity between the two treatment groups was previously tested and reported (Psomiadis et al., 2023, 2025). In brief, the two groups consisted of eight males and ten females each that were of similar age (mean at pre-treatment: 23.3 ± 7.8 years) and had similar facial convexity (mean facial contour angle at pre-treatment: 21.2 ± 4.9°) and overjet (mean at pre-treatment: 8.2 ± 2.5 mm). They received treatments of similar duration (mean: 2.8 ± 1.0 years), they had similar overjet at post-treatment (mean at post-treatment: 3.7 ± 1.7 mm) and differed only regarding the change of the facial contour angle induced by treatment (Camouflage: −1.2 ± 2.1°, Surgery: −6.2 ± 3.9°; Mann-Whitney U test p < 0.001).

Facial photographs

Originally, the photographs were obtained with the camera positioned at a standardized distance of 1.2 to 1.5 m from the subject’s face. Each image was edited to ensure uniform characteristics, namely a white background with uniform brightness and contrast, and standardized vertical facial dimensions (Version 22.0.1; Adobe Photoshop, San Jose, CA, USA, Adobe Systems). The Nasion–Menton distance was consistently set at 5 cm to eliminate size as a confounding factor in the assessments. Distinctive marks (e.g., moles, scars) and external elements (e.g., earrings, tattoos) were removed as well.

As published previously, following image adjustment, a configuration of four images per patient, consisting of pre- and post-treatment profile and frontal facial photos, was set in a landscape-oriented A4 size page (Psomiadis et al., 2023). A similar set of only facial profile photos was also created (Psomiadis et al., 2025). Both aforementioned sets were used to rate changes in facial appearance and the outcomes have been previously published (Psomiadis et al., 2023, 2025). For this study, patient images were displayed as shown in Fig. 1, to assess facial attractiveness.

Figure 1 Patient images, as displayed for rating of pre-treatment (A) facial profile attractiveness and (B) facial attractiveness.

Rater groups

Each set of images was evaluated by four groups of raters: (a) orthodontists, (b) oral and maxillofacial surgeons, (c) convex profile patients and (d) laypeople, aiming to obtain 10 ratings of each patient by members of each of the first three groups, and 20 by laypersons. Each rater assessed 12 individuals for each outcome (24 in total) to avoid fatigue (Gkantidis et al., 2013; Tsiouli et al., 2017; Zouloumi et al., 2019). The raters first completed twelve questionnaires assessing perceived changes in facial appearance resulting from treatment, as previously reported (Psomiadis et al., 2023, 2025). Immediately after this, they filled out another twelve questionnaires evaluating facial attractiveness, which concerned different individuals than those evaluated for facial appearance changes. The profile-related outcomes were rated in two consecutive sessions, while the overall facial outcomes were rated in another set of two consecutive sessions, with a gap of over 2 months between the two sets of rating sessions.

For this purpose, the patients under evaluation were randomly divided (https://www.random.org/) into three groups of twelve, with balanced sex distribution. Finally, each patient set was evaluated by 30 orthodontists, 30 oral and maxillofacial surgeons, 30 Class II patients, and 60 laypersons for changes in facial appearance following the two treatment approaches, as well as for pre-treatment facial attractiveness. The raters were not familiar with the patients being assessed, were unaware of the study’s aims, and did not know that the individuals rated were treated patients. Additional details on the composition of the rater groups are available in prior publications (Psomiadis et al., 2023, 2025).

Questionnaire fulfillment

The questionnaires used to assess changes in overall facial and profile appearance, along with the corresponding data reanalyzed in this study, were previously published (Psomiadis et al., 2023, 2025). For the evaluation of facial and profile attractiveness, each rater sequentially assessed 12 sets of photographs (Fig. 1) using a validated (Tsiouli et al., 2017; Zouloumi et al., 2019) printed questionnaire, which included a single question and a 100 mm visual analog scale (VAS) (Fig. 2). A total of 3,600 such questionnaires were administered by two calibrated researchers in a quiet room, with assessments conducted under discreet supervision.

Figure 2 The administered questionnaires to assess (A) facial profile attractiveness and (B) facial attractiveness at pre-treatment, through a 100 mm visual analogue scale (VAS).

Measurement recording

The distance from the left end of the VAS to each mark made by the raters was measured in millimeters with two decimal places with an electronic digital caliper (Jainmed, Seoul, Korea) and was recorded (Microsoft Excel spreadsheet, Microsoft Corporation, Redmond WA, USA). The error of the VAS rating measurements has been assessed previously and proved acceptable (Psomiadis et al., 2023).

Statistical analysis

Statistical analysis was performed using IBM SPSS Statistics for Windows (Version 29.0; IBM Corp, Armonk, NY, USA). Variance homogeneity was assessed with Levene’s test, while data normality was examined using the Shapiro-Wilk test along with Q-Q plots and histograms. Parametric statistics were applied.

Each patient received ratings from ten members of each rater group and 20 laypeople; the median of these ten ratings was used for further analysis as a reliable group assessment.

The agreement among different rater groups was evaluated using the intraclass correlation coefficient (ICC; two-way mixed model, absolute agreement, average measures). ICC values above 0.7 were interpreted as strong agreement. This analysis, along with comparative statistics across rater groups, supported the concurrent and statistical conclusion validity of the questionnaires.

Differences in facial profile attractiveness and in facial attractiveness between treatment groups and according to patient’s sex or rater group, as well as their interactions, were tested through univariate anaylsis of variance (ANOVA) (general linear model, all factors were fixed). Where applicable post-hoc analysis was performed through Fisher’s least significant difference (LSD) test.

Multivariate analysis of covariance (MANCOVA) was used to examine the effect of facial attractiveness on perceived changes in facial appearance. The raters’ responses served as the dependent variables, while the two treatment groups, the four rater groups, and the attractiveness ratings (as a covariate) were the independent variables. Patient’s sex was not included in the model following preliminary testing and according to previously published findings (Psomiadis et al., 2023, 2025). If significant results were found, separate ANOVAs would be conducted for each dependent variable, followed by post-hoc analysis using Fisher’s least significant difference (LSD) test. Two similar MANCOVA models were implemented, one using the facial attractiveness ratings and the other using the facial profile attractiveness ratings as a covariate. A power analysis for these models, which represent the main study outcomes, indicated that a sample size of 32 would be sufficient to achieve 80% power for detecting a medium effect size (Cohen’s f2 = 0.15) (G*Power software, version 3.1.9.2) (Faul et al., 2007). Consequently, the sample size of 36 was considered adequate.

A two-sided significance test was conducted in all analyses with an alpha level of 0.05. For pairwise post-hoc multiple comparisons, a Bonferroni correction was applied.

Results

Facial attractiveness outcome

Interrater agreement on facial attractiveness ratings, both overall and within each treatment group, ranged from good to excellent across all assessments (ICC > 0.80, Table 1, Fig. 3). Consistent variances of all dependent variables were detected across groups (Levene’s test, p > 0.05).

Table 1 Interrater agreement of VAS facial attractiveness ratings among groups of raters tested through the intraclass correlation coefficient (ICC; two-way mixed model, absolute agreement, results regarding average measures).

Treatment group	Frontal & Profile	Profile	
Total	0.88 (0.79, 0.93)	0.90 (0.81, 0.95)	
Camouflage	0.90 (0.80, 0.96)	0.91 (0.80, 0.96)	
Surgery	0.80 (0.59, 0.92)	0.88 (0.70, 0.96)	

Figure 3 Box plots showing the assessed (A) facial profile attractiveness and (B) facial attractiveness at pre-treatment condition, in VAS values (y-axis), grouped by rater type and by patient sex.

The upper limit of the black line represents the maximum value, the lower limit the minimum value, the boxed the interquartile range, and the horizontal black line the median value. Outliers as shown as black circles (±1.5IQR) or asterisks (±3IQR). The horizontal dashed line is placed at VAS value fifty, indicating a neutral attractiveness assessment.

Considering overall facial attractiveness, there were significant differences between the treatment groups at pre-treatment, without significant effects of patient’s sex, rater group, or any interaction on the outcomes (Table 2). The raters assessed the overall facial attractiveness at lower than average levels (mean: 36.1; 95% CI [33.7–38.5]), with patients treated exclusively by orthodontics found significantly more attractive than patients that received surgery (mean: 40.2; 95% CI [36.9–43.6] vs mean: 32.0; 95% CI [28.6–35.3], respectively) (mean difference: 8.2; 95% CI [3.5–13.0]; p < 0.001). Female patients were rated as slightly less attractive, but the differences from male patients were not statistically significant (mean difference: −3.6; 95% CI [−8.4 to 1.1]; p = 0.129). Differences between rater groups were small and non-significant, apart from one comparison between orthodontists and patients, with the latter providing slightly more positive assessments (mean difference: −7.9; 95% CI [−14.6 to −1.2]; p = 0.021) (Tables S1 and S2, Fig. 3B).

Table 2 Results of the ANOVAS testing the effect of patient’s sex, rater type, and treatment group on the facial and the facial profile attractiveness at pre-treatment condition.

Dependent variable	Factor	df	F	Sig.	
Facial profile attractivenessa	Treatment group	1	14.4	<0.001	
Sex	1	1.55	0.215	
Rater type	1	3.71	0.013	
Treatment group * Sex	1	0.50	0.483	
Treatment group * Rater type	3	0.66	0.579	
Sex * Rater type	3	0.55	0.651	
Treatment group * Sex * Rater type	3	0.06	0.982	
Facial attractivenessb	Treatment group	1	11.87	<0.001	
Sex	1	2.33	0.129	
Rater type	1	1.86	0.139	
Treatment group * Sex	1	0.27	0.604	
Treatment group * Rater type	3	0.21	0.886	
Sex * Rater type	3	0.05	0.984	
Treatment group * Sex * Rater type	3	0.35	0.792	
Notes:

a R Squared = 0.21 (Adjusted R Squared = 0.11).

b R Squared = 0.15 (Adjusted R Squared = 0.05).

df, degrees of freedom; F, F-value; Sig., Significance shown as p-values.

Regarding the facial profile attractiveness, significant between-group differences were evident at pre-treatment, without significant effects of patient’s sex or any interaction on the outcomes (Table 2). The facial profile attractiveness was rated at lower than average levels (mean: 37.7; 95% CI [35.5–40.0]), with patients treated exclusively by orthodontics found significantly more attractive than patients that received surgery (mean: 42.1; 95% CI [38.9–45.3] vs. mean: 33.4; 95% CI [30.2–36.6], respectively) (mean difference: 8.7; 95% CI [4.2–13.3]; p < 0.001). Female patients were rated as slightly less attractive than male patients, but not statistically different (mean difference: −2.9; 95% CI [−7.4 to 1.7]; p = 0.215). The only significant differences between rater groups were that patients’ ratings were significantly more positive than that of both specialists’ groups, but remained below average attractiveness (Tables S1 and S2, Fig. 3A).

Effect of facial attractiveness on perceived changes in facial appearance

For both MANCOVA models, the variances of all dependent variables were equal across groups (Levene’s test, p > 0.01).

Multivariate analysis detected a significant effect of pre-treatment facial attractiveness on perceived changes in facial appearance (F = 4.06, p = 0.002, Pillai’s Trace = 0.13, partial η2 = 0.13). There were no significant differences among rater groups (F = 1.35, p = 0.169, Pillai’s Trace = 0.14, partial η2 = 0.05), as well as no combined effects between rater group and treatment group on the outcomes (F = 0.78, p = 0.694, Pillai’s Trace = 0.09, partial η2 = 0.03). On the contrary, there were significant differences in perceived changes in facial appearance between treatment groups (F = 11.15, p < 0.001, Pillai’s Trace = 0.30, partial η2 = 0.30). Separate ANOVAs for each dependent variable showed findings consistent with the multivariate analysis in all cases (Table S3). Parameter estimates indicated that for one unit increase in the pre-treatment facial attractiveness there is an approximately 0.3 units decrease in the perceived changes in facial appearance in (β = −0.29, p < 0.001), with similar effects for all questionnaire items (Table S4). When accounting for the initial facial attractiveness of the patients, the raters still observed substantial positive changes in facial appearance among individuals who underwent a combined orthodontic treatment and orthognathic surgery. In contrast, minimal to negligible alterations in facial appearance were noted for those patients who exclusively received orthodontic treatment (Table 3).

Table 3 Estimated marginal means of perceived changes in facial appearance per treatment group and associated pairwise comparisons, accounting for initial facial attractiveness.

Dependent variable	Treatment group	Meana	Std. error	95% confidence interval	Mean difference (Surgery-Camouflage)	Std. error	Sig.	
Lower bound	Upper bound	
Face	Camouflage	52.01	1.50	49.04	54.98	14.23	2.17	<0.001	
Surgery	66.24	1.50	63.27	69.21	
Lower face	Camouflage	50.86	1.55	47.79	53.93	16.80	2.24	<0.001	
Surgery	67.66	1.55	64.59	70.72	
Upper lip	Camouflage	49.72	1.52	46.71	52.73	13.41	2.20	<0.001	
Surgery	63.13	1.52	60.12	66.14	
Lower lip	Camouflage	52.27	1.66	48.98	55.56	15.38	2.40	<0.001	
Surgery	67.65	1.66	64.36	70.94	
Chin	Camouflage	52.81	1.72	49.41	56.20	14.83	2.48	<0.001	
Surgery	67.64	1.72	64.24	71.03	
Notes:

Sig., Significance shown as p-values.

a Covariates appearing in the model are evaluated at the following values: Facial attractiveness = 35.9.

Multivariate tests detected a marginally not significant effect of the pre-treatment facial profile attractiveness on the perceived changes in facial profile appearance (F = 2.05, p = 0.075, Pillai’s Trace = 0.07, partial η2 = 0.07). There were small but significant differences among rater groups (F = 2.12, p = 0.009, Pillai’s Trace = 0.22, partial η2 = 0.07) and substantial significant differences between treatment groups (F = 16.32, p < 0.001, Pillai’s Trace = 0.38, partial η2 = 0.38), but no combined effects of these factors on the outcomes (F = 0.66, p = 0.825, Pillai’s Trace = 0.07, partial η2 = 0.02). Separate ANOVAs for each dependent variable detected significant effects of the pre-treatment facial profile attractiveness on the appearance of the face and the lower face, and marginally non-significant effects for the upper lip, the lower lip and the chin. There were also significant effects of treatment group factor, but non-significant effects of the rater factor, on all outcomes (Table S5). Parameter estimates indicated that for one unit increase in the pre-treatment facial profile attractiveness there was an approximately 0.2 units decrease in the perceived changes in facial appearance in (β = −0.20, p = 0.004), with similar trends for all questionnaire items (Table S6). When accounting for the initial facial profile attractiveness of the patients, considerable positive changes in the facial profile appearance of individuals that were subjected to orthognathic surgery were still perceived the raters. On the contrary, only minor changes were perceived for patients that received exclusively orthodontic treatment. This was evident for all outcomes apart from the upper lip (Table 4).

Table 4 Estimated marginal means of perceived changes in facial profile appearance per treatment group and associated pairwise comparisons, accounting for initial facial profile attractiveness.

Dependent variable	Treatment group	Meana	Std. error	95% confidence interval	Mean difference (Surgery-Camouflage)	Std. error	Sig.	
Lower bound	Upper bound	
Face	Camouflage	52.44	1.28	49.91	54.97	15.75	1.86	<0.001	
Surgery	68.19	1.28	65.66	70.72	
Lower face	Camouflage	53.07	1.38	50.33	55.80	16.13	2.01	<0.001	
Surgery	69.20	1.38	66.47	71.93	
Upper lip	Camouflage	48.87	1.40	46.10	51.65	13.02	2.04	<0.001	
Surgery	61.89	1.40	59.12	64.67	
Lower lip	Camouflage	53.33	1.55	50.27	56.39	14.55	2.25	<0.001	
Surgery	67.89	1.55	64.83	70.95	
Chin	Camouflage	53.49	1.42	50.68	56.29	16.69	2.06	<0.001	
Surgery	70.18	1.42	67.37	72.98	
Note:

Sig., Significance shown as p-values.

a Covariates appearing in the model are evaluated at the following values: Facial profile attractiveness = 37.6.

Discussion

This study examined the impact of pre-treatment facial attractiveness of convex profile patients with increased overjet on perceived changes in facial appearance, comparing outcomes from orthodontic camouflage vs. combined orthognathic and orthodontic treatment. The findings suggested that patients with lower pre-treatment facial attractiveness experience greater perceived improvement in appearance with the combined orthognathic and orthodontic treatment compared to those with higher initial attractiveness. Raters assessed the pre-treatment profile as well as the overall facial attractiveness of convex profile individuals at below average levels (36 and 38/100 VAS units, respectively), with minimal differences between rater types. This confirms the esthetic disturbance caused by the Class II facial configuration and reflects the associated treatment need (Pachêco-Pereira et al., 2015, 2016; Gabriele et al., 2024). Despite the pre-treatment group similarity indicated by standard dental and facial soft-tissue measurements, the initial profile and the overall facial attractiveness of patients treated with orthognathic/orthodontic treatment was rated 8 and 9/100 VAS units lower than that of orthodontics-only patients, respectively. This, along with the confirmation of our study hypothesis, highlights the need to control for the facial attractiveness factor when assessing the perceived treatment effects on facial appearance. This study confirmed the significant positive effects of combined orthognathic and orthodontic treatment on the profile and overall facial appearance of convex profile patients, with little to no improvement observed in cases treated with orthodontics alone, after controlling for initial facial attractiveness. This solidifies our previous findings that did not account for this factor (Psomiadis et al., 2023, 2025), and aids the decision making process, by indicating an enhanced benefit from treatment in cases of reduced initial facial attractiveness. The latter factor had a similar impact on the outcomes of both treatment types.

Taken all the above together, as well as the substantial evidence on the impact of facial appearance on several important life outcomes (Little, Jones & DeBruine, 2011; Kanavakis et al., 2021; Conville, Benson & Flett, 2024), it becomes apparent that convex profile patients with increased overjet and significantly reduced facial attractiveness should be oriented towards the combined orthodontic/orthognathic surgery approach. Under this prism, female patients might have slightly higher treatment need, since they were consistently rated as marginally less attractive that their male counterparts. However, given that this approach involves major surgery, patients should be thoroughly informed about the associated risks and potential morbidity (Hågensli, Stenvik & Espeland, 2014; Friscia et al., 2017; Agbaje, Luyten & Politis, 2018; Almasri et al., 2024). A less invasive alternative may be orthognathic camouflage procedures, which are not associated with the same level of morbidity or prolonged recovery as conventional orthognathic surgery (Burk et al., 2022b). This intermediate approach was not examined in the present study and warrants consideration in future research.

The digital photographs that were used for this comparison is a reproducible and valid tool for assessing dental and facial attractiveness (Todorov, Pakrashi & Oosterhof, 2009), and therefore, the most common method for this assessment. Slightly modified, entire facial photos (Tauk et al., 2016) of pre- and post-treatment conditions were presented simultaneously to the raters at random treatment status order and without disclosing any information to the raters (Ng et al., 2013). Through this approach, and by asking to assess changes in facial appearance, we aimed to minimize the effects of several factors that affect facial appearance judgments (Little, Jones & DeBruine, 2011; Kanavakis et al., 2021; Gabriele et al., 2024), while presenting realistic images to the raters (Hockley et al., 2012; Sari-Rieger & Rustemeyer, 2015). Thus, the evaluations concerned primarily changes in facial morphology due to the treatments being tested, without guiding the raters towards noticing any specific element. We believe this approach reduces the impact of confounding variables, thereby enhancing the ability to identify treatment effectiveness within the framework of personalized outcomes.

In the present sample of Class II patients, the pre-treatment facial attractiveness had a significant effect on the perceived changes in facial appearance post-treatment. The higher the pre-treatment facial attractiveness, the lower the potential improvement in facial appearance. This is in agreement with findings from the literature regarding orthognathic surgery patients, not only in terms of outcome assessment (Proffit, Phillips & Douvartzidis, 1992; Mihalik, Proffit & Phillips, 2003; Raposo et al., 2018), but also in terms of patient satisfaction and decision to seek treatment (Vargo, Gladwin & Ngan, 2003; Pachêco-Pereira et al., 2016). Previous studies also suggested that the chances of significant improvement in facial appearance of convex profile patients following orthognathic surgery are higher when the initial sagittal skeletal discrepancy surpasses a cephalometric threshold (Shelly et al., 2000; Tsang et al., 2009). However, it should also be considered that larger changes may carry a greater risk of relapse, which might lessen these effects over the long term (Mihalik, Proffit & Phillips, 2003). Facial attractiveness levels have also been shown to significantly influence the perceived benefits of smile enhancement following orthodontic treatment, with a greater positive impact of posttreatment smile in less attractive females (Farshidnia et al., 2023). The facial outcome is also mediated by other facial features, such as the nose (Raposo et al., 2018; Cankaya, Celebi & Bicakci, 2019). Therefore, substantial evidence that facial attractiveness has a key role in decision making and impacts the perceived outcomes related to patient satisfaction was already available. The present study directly assessed this factor and quantified its impact, solidifying the previous evidence. It also confirmed that the substantial benefits in facial appearance detected after combined orthodontic/orthognathic surgery treatment remain clearly superior than that of the orthodontic treatment alone after controlling for the initial facial attractiveness factor. Minimal changes were demonstrated in facial appearance for those patients who exclusively received orthodontic treatment. Therefore, it can be argued that the orthognathic surgery approach should be the first treatment of choice in all convex profile individuals that seek improvement of their facial appearance. The facial attractiveness factor should be considered in decision making, among several other factors, as a factor that mediates treatment benefits and should be communicated to patients as such. The amount of this effect has been determined at 0.3/100 VAS units decrease in the perceived changes in facial appearance for every 1 unit increase in overall facial attractiveness. The corresponding effect on profile assessment was reduced at 0.2/100 VAS, probably due to restricted facial view, which confounded attractiveness assessments (Liu et al., 2024). This highlights the important of assessing treatment effects on different facial views, and not only in profile aspects, since people present themselves from various viewing angles during daily interactions (Gkantidis et al., 2013; Zouloumi et al., 2019; Psomiadis et al., 2025). Therefore, we believe that the simultaneous frontal and profile assessments and the associated findings are more clinically relevant and should be primarily considered during treatment planning and decision-making.

Overall, the different rater groups assessed similarly the patients’ pre-treatment facial attractiveness, as was the case for treatment effects on facial appearance (Psomiadis et al., 2023, 2025). This enhances the generalizability of the findings in terms of treatment benefits related to facial appearance during human interactions. Regarding facial attractiveness scores, there were only few small differences towards a more positive assessment by the patients’ group as compared to specialists’ groups, which were not considered important. Other studies, not directly comparable to ours, showed contradictory findings in this matter (Ng et al., 2013; Gkantidis et al., 2013; Sari-Rieger & Rustemeyer, 2015; Reis et al., 2021).

Among the strengths of the present study is the use of validated questionnaires on the respective population (Tsiouli et al., 2017; Zouloumi et al., 2019), the exclusion of outliers from the sample in terms of facial morphology, the pre-treatment group similarity, the balanced demographic and morphological characteristics of the treatment groups, the diverse groups of raters, including laypeople, the slightly modified entire facial photos, the simultaneous presentation of pre- and post-treatment statuses in random order, and the use of printed questionnaires that were fulfilled at standardized conditions. Furthermore, the multivariate analysis models allowed for simultaneous assessment of several factors and their interactions adding validity to the outcomes and enhancing the study findings interpretation and generalizability.

A limitation of the study is that facial attractiveness, as well as treatment outcomes regarding facial appearance, were assessed statically and functional aspects, such as the smile, were not evaluated. Thus, a potential positive effect of orthodontic treatment alone in dental and smile aesthetics (Coppola et al., 2023; Farshidnia et al., 2023) was not captured. Therefore, the social benefits from orthodontic treatment alone might have been underestimated here. Moreover, the study included only white individuals. The findings may be modified in individuals with different skin color, as well as in individuals with more extreme facial morphology. The aforementioned factors need to be assessed in future research.

Conclusions

This study highlights the critical role of initial facial attractiveness in shaping perceived treatment outcomes for patients with convex facial profiles and increased overjet. Patients treated exclusively with orthodontics were rated as more attractive at baseline compared to those undergoing combined orthodontic and orthognathic surgery. However, after adjusting for initial attractiveness, the surgical group showed significantly greater perceived improvements in facial and profile appearance, particularly among patients with lower initial attractiveness.

Minimal differences among rater groups confirmed the consistency of attractiveness assessments. The inverse association between pre-treatment attractiveness and perceived improvement suggests that less attractive individuals experience greater benefits. Combined orthodontic/orthognathic treatment led to substantial positive changes, while exclusively orthodontic treatment showed minimal impact.

Clinicians should incorporate these findings into patient counseling and treatment planning, providing individualized information on expected outcomes based on initial facial attractiveness to support informed, evidence-based decisions.

Supplemental Information

Supplemental Information 1 Supplementary analyses of facial and profile attractiveness ratings and perceived treatment effects across rater groups in patients with convex facial profiles.

Detailed statistical analyses related to the evaluation of facial and profile attractiveness and the perceived changes following treatment of convex facial profiles. Tables S1 and S2 summarize attractiveness ratings and pairwise comparisons across rater groups (surgeons, orthodontists, patients, and laypeople). Tables S3 and S5 report ANOVA results testing the effects of initial attractiveness, rater type, and treatment modality (surgery vs. camouflage) on perceived changes in facial and profile appearance. Tables S4 and S6 provide parameter estimates quantifying the influence of these factors on specific facial features (face, lower face, lips, and chin).

Supplemental Information 2 Anonymized raw questionnaire data, including the questionnaire ratings used for all analyses.

Additional Information and Declarations

Competing Interests

Nikolaos Gkantidis is an Academic Editor for PeerJ.

Author Contributions

Simos Psomiadis conceived and designed the experiments, performed the experiments, analyzed the data, prepared figures and/or tables, authored or reviewed drafts of the article, and approved the final draft.

Ioannis Iatrou conceived and designed the experiments, authored or reviewed drafts of the article, and approved the final draft.

Iosif Sifakakis conceived and designed the experiments, performed the experiments, prepared figures and/or tables, authored or reviewed drafts of the article, and approved the final draft.

Nikolaos Gkantidis conceived and designed the experiments, analyzed the data, prepared figures and/or tables, authored or reviewed drafts of the article, and approved the final draft.

Human Ethics

The following information was supplied relating to ethical approvals (i.e., approving body and any reference numbers):

The study protocol received approval from the Research Ethics Committee of the Dental School at the National and Kapodistrian University of Athens, Greece, prior to its initiation (Approval Date: 22.06.2018, Protocol Number: 361).

Data Availability

The following information was supplied regarding data availability:

The anonymized raw questionnaire data, including the questionnaire ratings used for all analyses, are available in the Supplemental Files.

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
