# Peer review of "Role of initial facial attractiveness in the perceived aesthetic outcome of convex profile treatment"

_PeerJ, doi:10.7717/peerj.19997_

## Round 0.1 · original submission · Major Revisions

Please justify how this research differs from/builds upon your previous publications.

·

Basic reporting

Clear and professional English formatting used throughout the manuscript.

Experimental design

Methods and material described in detail.Though similar research published previously,the extent of the study is increased by including 3600 ratings to make the results more robust.

Validity of the findings

No comments

·

Basic reporting

STROCCS criteria adequacy should be included.

Experimental design

Starndardised distance for Facial photographs should be mentioned.

Validity of the findings

it is not clear which kind of surgical orthognathic procedure patients underwent. Orthognathic surgery is general term and the kind of procedure ( LF1, BSSO, CCWR, genioplasty) as well as type of bone replacement ( advancement for instance) in one or both jaws should be specified and discussed since it can influence the final results.
It would be interesting to discuss if the authors have stratified results according to the kind of surgical procedure.

Additional comments

the present paper should be cited: doi: 10.1007/s12663-024-02225-1

Reviewer 3 ·

Basic reporting

There are several instances throughout the manuscript where language could be improved for clarity and correctness. In particular, there are noticeable issues at lines 115, 313, and 341, including grammatical mistakes and awkward phrasing. I recommend a thorough proofreading or language editing by a native speaker or professional service.

Experimental design

The current study is titled "Impact of Facial Attractiveness on Perceived Treatment Outcomes in Convex Profile Patients." However, the authors should more clearly differentiate this work from their previous publication, "Perceived Effects of Orthognathic Surgery versus Orthodontic Camouflage Treatment of Convex Facial Profile Patients."
It would be beneficial for the authors to explicitly state:

What novel research question is being addressed in the current study.

How the methodology or perspective differs from the prior work.

What added value or new insights this study brings to the field.

Validity of the findings

Previous studies by the same group have primarily focused on the comparison between orthodontic camouflage and orthognathic surgery. Given that the present study emphasizes the role of facial attractiveness in perceived treatment outcomes, the conclusions should be better aligned with this central theme.

·

Basic reporting

1 I suggest revising the title to clearly indicate the type of study (e.g., retrospective, clinical trial) and specify what is being investigated. For instance, a more descriptive title might be: "A Comparison Between Orthognathic Surgery Combined with Orthodontics and Orthodontic-Only Treatment in Perception of Facial Esthetics." This approach helps the reader understand the scope of the study from the outset.
2 Zymperdikas and Koresty are systematic reviews that focus on cephalometric standards rather than the impact on facial profile. In contrast, Liu et al. (2024) demonstrated notable improvements in facial profile following orthodontic treatment. Additionally, Shell (2003) presents findings that do not support the statement made in line 60. Regarding line 60: the authors have already addressed this topic in previous studies, noting a slight yet noticeable improvement. Therefore, the phrase "remains unproven" may not accurately reflect the current evidence. The contributions of Liu and Shell should be considered when discussing this point, particularly in lines 59 and 60.
3 Including the studies by Mihalik et al. (2003; DOI: 10.1067/mod.2003.43) and Raposo et al. (2018; DOI: 10.1016/j.ijom.2017.09.003) would enhance both the background and discussion sections by providing additional context and supporting evidence.

Experimental design

The central research question appears to be whether there is a difference in perceived attractiveness before and after treatment in patients undergoing either orthodontic camouflage or combined surgical treatment. Although the study is retrospective, the observed differences in initial attractiveness ratings suggest that the groups were not comparable at baseline. This disparity complicates direct comparisons, as profiles perceived as more attractive at the outset are naturally subject to less noticeable changes.

While the study does provide insights into how attractiveness may influence treatment indication, this was not the stated primary research question. To directly address the intended question, it would be necessary to ensure that both treatment groups are matched not only in terms of dental Class II diagnosis but also in skeletal and attractiveness characteristics. This would allow for a more meaningful comparison between patients who could plausibly be treated with either orthodontic camouflage or orthognathic surgery.
Furthermore, the two previous studies by the authors (2023 and 2024) appear to be quite similar in design and scope. The primary difference seems to be the inclusion of both frontal and profile views rated on the same scale, whereas the earlier study evaluated only profile views. However, combining frontal and profile views in a single evaluation may influence rating outcomes, as the perception of attractiveness can differ significantly depending on the angle of view.

Validity of the findings

The study presents valid results and appears to have been conducted with methodological care. However, it closely mirrors the authors' two previous publications, with only minor variations in design—namely, the combined analysis of frontal and profile views. As a result, the current manuscript offers limited novelty and does not substantially advance the findings already presented in the earlier studies.

Additional comments

A potential improvement would be to assess the frontal and profile views separately. One suggested approach is to randomly mix baseline and final images—both frontal and profile—and have raters evaluate attractiveness without knowing whether each image represents a pre- or post-treatment condition. This would reduce bias and provide a clearer understanding of how each view contributes to perceived changes.

Additionally, given the retrospective nature of the study, it might be feasible to invite these patients for a follow-up evaluation to assess long-term changes in attractiveness, which would add depth and novelty to the current findings.

As it stands, the current study closely mirrors the previous work, making it challenging to justify publication without more distinct methodological advancements or deeper analysis.

---

## Round 0.2 · accepted · Accept

I have assessed the revised manusript. The authors have adequately addressed the comments of the referees. The applied statistics are adequate.
The manuscript is ready for publication now.

·

Basic reporting

The authors have improved the paper according to the review comments.

Experimental design

-

Validity of the findings

-